# A Fundamental Study on a Porous Carbon Nanotubes Macroelectrode in Weakly Supported Electrolyte: A Novel Criterion for Distinguishing Diffusion Domains

**DOI:** 10.3390/ijms26178262

**Published:** 2025-08-26

**Authors:** Josipa Dugeč, Ivana Škugor Rončević, Nives Vladislavić, Marijo Buzuk

**Affiliations:** Department of General and Inorganic Chemistry, Faculty of Chemistry and Technology, University of Split, 21000 Split, Croatia; josipa.dugec@ktf-split.hr (J.D.); skugor@ktf-split.hr (I.Š.R.); nives@ktf-split.hr (N.V.)

**Keywords:** porous electrode, SWCNTs, mass transport domains, reverse pulse voltammetry, double potential step chronoamperometry, double-layer effects

## Abstract

A new approach is presented to elucidate the phenomena that occur within a porous single-walled carbon nanotubes (SWCNTs) modified glassy carbon electrode (GCE) and that influence the electrochemical behavior of the modified electrode. By employing cyclic voltammetry, reverse pulse voltammetry, and double potential step chronoamperometry, insights into the structural changes in the electrochemical double layer and the mass transport regimes are gained. An analysis of the reduction of the electrochemically generated [Fe(CN)_6_]^3−^ shows that the SWCNTs layer can be considered inactive. However, their pronounced influence on the electrochemical signal arises from their capacitive behavior. Furthermore, a novel criterion for distinguishing the mass transport domains is proposed, which allows the estimation of the points at which a change in the mass transport regime occurs. The results also show the role of the porous SWCNTs layer in preventing the expansion of the double layer as well as in the process of ion condensation in the Gouy-Chapman layer. Finally, the counterintuitive and unexpected voltametric behavior, such as the independence of the current peak heights from the ionic strength of the support, the parabolic dependence of the peak potential on the scan rates, and the occurrence of steady-state currents, are discussed.

## 1. Introduction

Since the discovery of single-walled carbon nanotubes (SWCNTs) by Iijima [1] and Ichihashi [2], these remarkable materials have been used in various electrochemical applications, with particularly promising prospects for electrical energy storage and the development of (bio)sensors.

In general, electrodes modified with a SWCNT layer show a significantly improved electrochemical response compared with their unmodified counterparts. Although there is no doubt about the dependence of the electrical properties on the structure of SWCNTs [3], there is some uncertainty about the reasons for the improved electrochemical response of SWCNTs. The role of the basal and edge planes of CNTs [4,5,6], the oxygen-containing functional group on CNTs [7,8], as well as the influence of metallic impurities [9,10,11] in heterogeneous electron transfer on carbon nanomaterials, has been investigated by Compton research groups.

Also, due to their porous morphology, the mass transport phenomena in the CNTs layer exhibit a complex behavior that affects the overall electrochemical response in a way that can easily be misinterpreted as “electrocatalytic”. This is not surprising since the mass transport phenomena influence the peak-to-peak separation, the lower overpotential of the electrochemical response, etc. [12,13,14]. Related to this, various approaches have been reported to explain the enhanced electrochemical response of electrodes covered with porous layers. These studies included voltametric and chronoamperometric simulations and experimental investigations of various ordered porous geometries (pinholes, cylinders, spheres, etc.), which are assumed to be electrochemically active, partially active, or insulating. The results of these studies are summarized in our recent article [15]. Shortly, the significant contribution of thin-layer effects to the cyclic voltametric signals of nicotine on MWCNTs was observed by Sims et al. [16]. Carrara et al. investigated the role of SWCNTs in biosensing systems, analyzing various electrochemical effects (Nernst, Randles–Sevcik, and Cottrell effects) [17]. The electrochemical behavior of some common redox couples or electroactive species, such as Ru(NH)_3_^3+/2+,^ on different carbon nanomaterials [18] and the analysis of the electron transfer kinetics of Fe(CN)_6_^4−^ by cyclic voltammetry on a porous layer prepared by drop-casting of SWCNTs [19] were also reported.

Recently, Kinkelin et al. [20] extended the findings of Punckt et al. [21,22] regarding the explanation of the phenomena that determine the formation of CV peaks in porous media. This study focuses on distinguishing the current signal generated in a porous CNTs/Nafion^®^ layer on a GCE from the signal originating from the surface of the modified electrode layer. A simulation model based on the Butler–Volmer equation and Fick’s law of diffusion was also presented. As a conclusion, this subject attracted the attention of other researchers and led to several provocatively titled studies [17,23,24,25].

Traditionally, a high concentration of supporting electrolyte is employed in electrochemical experiments in order to obtain quantitatively diffusional conditions. The influence of ionic strength on the electrochemical responses of myoglobin loaded in polysaccharide assembly films was shown in the study of Liu et al. [26]. Dickinson et al. investigated how much supporting electrolyte is required to make a cyclic voltammetry experiment quantitatively diffusional in the case of [Ru(NH_3_)_6_]^3+/2+^ at a Pt macroelectrode [27]. The role of the supporting electrolyte cation and the impact of a low supporting electrolyte concentration on the kinetics of outer-sphere Cr(OH_2_)_6_^3+^ and Cr(NH_3_)_6_^3+^ complex, was investigated by Weaver et al. [28]. The findings on the influence of the structure and distribution of the ionic species in the electrochemical double layer on the heterogeneous electrochemical reaction were also summarized by Dunwell et al. [29].

The behavior of diffusion layers in systems with slow charge transfer has been studied by Molina and co-workers [30]. Finally, besides the above-mentioned phenomena, two significant effects, especially in the case of a nanoelectrode [31], must be taken into account when the experiments are performed at low support: the Levich effect and the Frumkin effect [32].

This paper aims to elucidate, from a mechanistic point of view, the counterintuitive and unexpected voltametric phenomena that occur at the porous electrode, especially in low-support electrolytes. As previously mentioned, despite numerous models and simulations on mass transport phenomena in porous electrodes, an explicit conclusion or even the occurrence of the characteristic phenomena predicted by simulation models is rarely reported in real systems. Moreover, due to the complex and disordered morphology of the drop-casted layers, most simulations and experimental investigations are limited to being applied to organized porosity systems with different geometries. Here, for the first time, we present the possibility of investigating electrochemical phenomena occurring in porous SWCNTs films that match (at least in some parts) those predicted in simulation studies for different organized geometries [15]. Accordingly, the results presented in this study serve as evidence for the occurrence of the phenomena predicted in the simulation study. An additional value of this article is that this relationship has been established between the experimental results obtained on an electrode with a highly amorphous morphology of the drop-casted SWCNTs layer and the results predicted by simulations for highly organized morphological systems. Accordingly, the observed effects, responsible for the deviation of the voltametric response from those expected by the Randles–Sevcik behavior, are investigated and critically discussed. For this purpose, voltammetry and chronoamperometry at different ionic strengths of the support were performed. As a result of this investigation, it was found that heterogeneous electron transfer occurs at a glassy carbon electrode. Furthermore, here, we propose a novel criterion that can be used in the differentiation of the two mass transport domains. Based on this criterion, the time and the slopes of the log(*I*) vs. log(*t*) dependence, which indicate a switch in mass transport regime, are estimated.

## 2. Results and Discussion

### 2.1. Cyclic Voltammetry (CV) Study

One of the standard approaches in determining the control mechanism of the electrochemical response is the influence of the different scan rates on the CVs. The recorded CVs at different electrodes and different ionic strengths of KCl at GCE are presented in the Appendix A, while the corresponding analysis of the obtained voltammograms can be seen in Figure 1. As shown in Figure 1a, the slopes of log(*i*) vs. log(*υ*) for oxidation indicate a diffusion-controlled mechanism, independent of KCl concentration. Some discrepancies are observed at higher scan rates for the lowest support. 

As expected, an increase in scan rate is followed by a shift in peak potentials towards more anodic values, and this effect becomes more pronounced with decreasing KCl concentration (Figure 1b). Interestingly, the discrepancies in the cathodic peaks are more pronounced when the measurements were performed at low support concentration (Figure 1c). This is followed by a shift in the peak potential toward more cathodic values, regardless of the ionic strength of KCl (Figure 1d). This was the first issue that caught our attention.

An identical analysis for the SWCNTs electrode is presented in Figure 2, while the recorded CVs can be seen in Appendix A. Surprisingly, the log(*i*) vs. log(*υ*) are almost the same for all conditions and peaks (Figure 2a,c). In addition, the current peak heights seem to be independent of the KCl concentration. This is followed by the unusual dependence of the anodic potential peak heights vs. scan rates (Figure 2b). This behavior was not observed for the dependence of the cathodic peak potential vs. scan rate. This was the second problem that attracted our attention. A detailed discussion of these results can be found in Section 2.4.

### 2.2. Double Potential Step Chronoamperometry (DPSCA) Study

Double potential step chronoamperometric (DPSCA) experiments were performed to gain insight into the processes taking place at the GCE modified with SWCNTs. This method was chosen due to our focus on studying the cathodic reaction, as this reaction should have fewer degrees of freedom, as it is independent of the concentration of electroactive species in the bulk. This means that the investigation of the electrochemical behavior can only be restricted to the porous SWCNTs layer if parameters such as the concentration of the Fe(II) and the concentration of the supporting electrolyte are chosen properly. This is based on the fact that the electroactive species—Fe(III)—only originates from the electrochemical oxidation of the initial Fe(II) solution. Therefore, the concentration variation of Fe(III) and, accordingly, the chronoamperometric response should be more pronounced in the cathodic branch of the DPSCAs. This concentration can be controlled by several parameters. The first parameter is the effective potential difference. In order to keep the degrees of freedom as low as possible, the variation of the effective potential difference was achieved by using a supporting electrolyte with different ionic strengths. In this way, the same potential steps can be used during the DPSCA measurements. The second parameter is the Fe(II) solution concentration. The details of the experimental conditions are given in the Materials and Methods.

The analysis of the experimental data shows that the most meaningful DPSCA responses and their derivatives, which provide the best understanding of the electrochemical phenomena in the porous SWCNT electrode, were noticed in 0.001 M KCl. The recorded DPSCAs, as well as the Cottrell analysis at different ionic strengths, are given in Appendix A. As shown, the given dependence does not provide sufficient evidence to draw relevant conclusions.

However, an interesting behavior can be observed when an appropriate analysis is performed. This analysis includes the representation *I*_total_/*I*_blank_ vs. *t* (Figure 3), log(*I*) vs. log(*t*) (Figure 4), d(log(*I*))/d(log(*t*)) vs. log(*t*) (Figure 5). The presentation in Figure 3 is based on our previous use of this analysis to determine the ‘apparent catalytic rate constant’ [33], which aims to elucidate mass transport phenomena within the porous layer. However, some changes have been made to the earlier presentation, since in this work, the dependence of *I*_total_/*I*_blank_ on *t*, but not on *t*^1/2^ as in the original equation by [34], is shown. *I_t_*_otal_ and *I*_blank_ represent the currents recorded in the presence and absence of the electroactive species, respectively. This simplification is made because the ratio *I*_total_/*I*_blank_ essentially represents the ratio between the pure faradaic current (*I*_farad_), which originates from the electrochemical reaction of the Fe(II) and Fe(III) species, and the “blank” current, i.e., the capacitive current (*I*_blank_), as follows:(1)$$\frac{I_{total}}{I_{blank}}=\frac{I_{farad}+I_{blank}}{I_{blank}}=\frac{I_{farad}}{I_{blank}}+1$$

As will be shown, this representation enables a more distinctive approach and offers a deeper insight into the phenomena that occur during the DPSCA measurements.

For the sake of clarity, the representations of log(*I*) vs. log(*t*) and d(log(*I*))/d(log(*t*)) vs. log(*t*) for the cathodic branch of DPSCAs are also adapted to the relative time from 0 s (10 s in DPSCAs) to 10 s (20 s in DPSCAs).

Since the cathodic response shows a more diverse behavior, the discussion of the analysis results will begin with the cathodic branch of DPSCAs. The *I*_total_/*I*_blank_ vs. *t* branch of the DPSCAs is shown in Figure 3a. The presented curve can be explained as follows. After switching the potential to –0.1 V, a double-layer charging current occurs, characterized by the decrease in the *I*_total_/*I*_blank_ ratio as the ionic strength of the supporting electrolyte decreases. This behavior is to be expected, as charging is particularly slow at a low ions concentration. The increase in the *I*_total_/*I*_blank_ ratio is then a consequence of the increase in faradaic currents. After the maximum value of the *I*_total_/*I*_blank_ ratio is achieved (peak), the *I*_total_/*I*_blank_ ratio begins to decrease. In the case of 0.001 M KCl, this ratio reaches the lowest value (*I*_total_/*I*_blank_ = 1, point “*I*_0_” in Figure 3b) at 17.8 s, indicating that no more Fe(III) is present in the system. In conclusion, the phenomena during the applied cathodic potential can be roughly divided into two regions.

In order to elucidate the phenomena occurring in these two regions, the dependencies log(*I*) vs. log(*t*) and d(log(*I*))/d(log(*t*)) vs. log(*t*) were examined in detail. At this point, it should be emphasized that all dependencies derived from the DPSCAs are based on faradaic currents. As presented in Figure 4b, a significant variation of the dependencies of log(*I*) vs. log(*t*) was obtained in the case of the cathodic branch of DPSCAs in 0.001 M KCl. As previously concluded from Figure 3b, the faradaic current disappears at 17.8 s (i.e., relative time in cathodic branch of DPSCAs of 7.8 s), which is consistent with the point “*I*_0_” (log(*t*) = 0.89) in Figure 4b.

Accordingly, under these conditions, the system needs 7.8 s to totally depletion of the Fe(III). This strongly suggests that SWCNTs are not involved as an active center (at least not to an extent that has a significant impact on the overall current signal) for heterogeneous electron transfer, as their large specific surface area and porosity are expected to behave similarly to thin-layer cells. Accordingly, the electrochemical reaction seems to take place on the surface of the GCE. This assumption is strongly supported by the analysis of Figure 5c at short times. As shown, the bell curve of d(log(*I*))/d(log(*t*)) vs. log(*t*) was obtained during short times for 0.001 M KCl (black curve). The ascending part of the curve indicates the formation of the concentration profile of Fe(III) in the vicinity of glassy carbon. After the formation of the concentration profile near the GCE, a short plateau around log(*t*) ≈ −0.1 can be observed, characterized by a minimum of the slope (about −0.2). This plateau reflects a steady-state current. This is possible at a low effective potential in 0.001 M KCl, as this condition favors the kinetic control of the electrochemical reaction. As the electrolysis proceeds, Cottrellian behavior occurs as a diffusion profile is formed and confined in the vicinity of the GCE. After this region, there is a sharp increase in the slope as the Fe(III) is hardly present in the SWCNTs layer. This is a consequence of the rapid depletion of the remaining Fe(III) in the SWCNTs layer. However, from this analysis, it is very difficult to distinguish when the SWCNTs layer is depleted of Fe(III). The boundary between the mass transport through the SWCNTs layer and where the mass transport through solution also occurs can be estimated by a comparative analysis of the dependencies presented in Figure 2b. The peak that occurs in *I*_total_/*I*_blank_ vs. *t* represents the threshold of these two regimes. At Fe(II) concentration of 0.1 mM and an ionic strength of 0.001 M KCl, the peak occurs at 11.3 s (point “a”). In Figure 4b and Figure 5c, it corresponds to the log(*t*) of 0.16 (note: the plot includes the relative cathodic time, i.e., 1.3 s), which is marked as point “a”. From the Figure 5c, the value of d(log(*I*))/d(log(*t*)) vs. log(*t*) was estimated as −0.63. Accordingly, the region between points “*I_0_*” and “a” can be characterized as the regime in which the change in flux also occurs through the solution.

Relied on the above, the cathodic branch of DPSCAs can be analyzed for higher KCl ion strengths. As shown in Figure 4b and Figure 5b,c, no significant deviation in dependences was observed, as it was noticed in the case of 0.001 M KCl. This can be attributed to a higher initial electrochemically generated Fe(III) amount in the system due to the higher effective potentials at higher KCl concentration. Furthermore, the higher effective potential difference has a significant influence on the reduction of Fe(III). This manifests itself in a lack of detection of the processes at short times. As shown in Figure 3b, at time 10 s (relative cathodic time 0), the applied potential is such that the Fe(III) present in the vicinity of GCE is depleted (linear part of cathodic branch up to point “b”) on a time scale of about 0.3 s, afterwards Cottrellian behavior occurs in the SWCNTs layer. Due to the interplay of the higher effective potential and the electrochemically generated amount of Fe(III), this process takes somewhat longer than in the case of 0.001 M KCl. The times and related parameters determined from Figure 3, Figure 4 and Figure 5 are listed in Table 1. As shown, the determined values of slopes (points “a”) are around −0.65, which confirms that this value can be used as a threshold between two regimes.

This analysis is simpler when applied to the anodic branches of DPSCAs. As can be seen from Figure 3a, the processes during the anodic branch of the DPSCAs take place mainly in the porous SWCNTs layer. This was to be expected as Fe(II) is initially present in the solution. As a consequence of the higher concentration in the bulk of the solution, the slope values tend toward Cottrellian behavior (approaching −0.45) at longer times (Figure 5a). Of course, some differences can be observed for short times, as the values of the slope is lower for higher KCl concentrations. As explained in the case of the cathodic DPSCA signal, the lower effective potential leads to the establishment of a local diffusion in the vicinity of the electrode. However, with a higher ionic strength of the supporting electrolyte, the depletion of Fe(II) in the vicinity of the GCE is more emphasized. This means that the diffusion layer is wider than in the case of the 0.001 M KCl. As a result of the above-mentioned, slower expansion of the diffusion layer at a given effective potential difference can be expected.

In a further experiment, DPSCA measurements were carried out at different Fe(II) concentrations in 0.001 M KCl. The obtained DPSCAs, as well as the Cottrell analysis, are included in the Appendix A, while the analysis of the cathodic branch of the DPSCAs is presented in Figure 6, Figure 7 and Figure 8. As can be seen from Figure 6a, the anodic branch of the DPSCAs shows the expected behavior with increasing Fe(II) concentration, as the peak is reached quickly at lower Fe(II) concentrations, indicating a slower expansion of the diffusion layer at higher Fe(II) concentrations. Accordingly, more information about the phenomena in the porous layer can be obtained from the cathodic branch of the DPSCAs.

Interestingly, similar behavior was observed at higher Fe(II) concentration, as in 0.01 M and 0.1 M KCl at 0.1 mM Fe(II) (Figure 7). These trends can be explained by the self-support effect at low support concentration and a relatively high amount of electroactive species, especially in the case of the small and limited volume in the vicinity of the GCE. This conclusion can be supported by the dependence of log(*I*) vs. log(*t*) at higher Fe(II) concentration in different KCl solutions. As presented in Appendix A, no significant difference can be observed in the cathodic branch of the DPSCAs. Another interesting point is the observation of the steady-state current on the short time scale for 0.2 mM Fe(II) after the slope starts to increase and reaches values around −0.5 on the same time scale as in the case of 0.1 mM Fe(II) (Figure 7 and Figure 8). This behavior is most similar to that predicted by Yang et al. [35] in analyzing the simulation of the dimensionless flux time transition to a planar electrode cover with different numbers of non-conducting layers. However, in contrast to the reported simulation, in our case, a rapid depletion in Fe(III) leads to a rapid decline of the slope with time. This happens due to the fact that it can be assumed that the concentration of the Fe(III) in the bulk of the solution is zero.

Additional evidence for the above thesis was found by enlarging the GCE surface area by grounding the GCE with paper 3000, after drop casting of SWCNTs was performed. DPSC experiments with this electrode (GCE-3000 SWCNTs) were performed at different concentrations of Fe(II) in 0.001 M KCl. Interestingly, no valuable data for the cathodic branch of the DPSCAs were obtained when the solution contained 0.1 mM Fe(II) (Figure 9). However, at 0.5 mM, a signal more similar to that recorded with a SWCNT electrode (polished GCE) at 0.1 mM Fe(II) was obtained (see Figure 10 and Figure 11). In summary, as the GCE surface area increases, the described phenomena can be observed at higher Fe(II) concentration. The diversity of the signal presented in Figure 11b is beyond the scope of this article, as the explanation for this dependence is to be found in the mass transport that takes place both at the tips and in the “valley” of the scratched GCE [36,37,38].

Finally, Figure 12 shows a comparison of the DPSCAs result obtained with polished GCE and SWCNTs electrodes. The DPSCAs for GCE were performed in the presence of 1 mM Fe, which ensures a sufficient concentration of Fe(III) near the electrode. A lower Fe(II) concentration did not result in a valuable cathodic signal. The behavior in the cathodic branch of the DPSCAs shows a remarkable similarity with the results recorded at lower Fe(II) concentrations at the electrode modified with SWCNTs. All this supports the assumption that GCE is an electroactive surface in this system.

### 2.3. Reverse Pulse Voltammetry (RPV) Study

Although the detailed analysis in the previous Section indicates that the electrochemical reaction takes place at the GCE, it cannot provide sufficient evidence to clarify the voltametric behavior. This is mainly a consequence of the fundamental differences between these two techniques. Accordingly, reverse pulse voltammetry (RPV) is harnessed in this study to investigate the phenomena that occur during voltammogram scans. RPV was chosen in relation to the information obtained from the chronoamperometric analysis. In principle, this technique can be considered as a series of DPSC experiments at different second step potentials if appropriate parameters are adapted to the system under study. In this work, RPV was performed on the SWCNTs electrode and GCE in the presence of 0.5 mM Fe(II). The experiments were performed at different KCl concentrations. For simplicity, the influence of the pulse time and KCl concentration at a scan rate of 50 mV s^−1^ for the SWCNTs electrode and GCE are presented.

Figure 13 shows reverse pulse voltammograms (RPVs) with a pulse time of *t*_p_ = 0.05 s for the SWCNTs electrode and GCE. As shown, in low support (Figure 13a), the SWCNTs electrode shows a continuous increase in cathodic current over the entire potential range. At low ionic strength of KCl, the charging currents are very high as the SWCNT layer acts as an electrochemical double-layer capacitor (EDLC). In combination with the short pulse time (*t*_p_ = 0.05 s), at anodic potentials, cathodic current is a consequence of the relaxation currents, while at more negative potentials, the charging currents dominate. This contribution is also expected to increase when the potential flows to more negative values due to a higher stimulus pulse. The voltammogram recorded at GCE in 0.001 M KCl will be commented on later (see discussion for Figure 13d). For comparison, RPVs recorded, for both electrodes, in 0.1 M KCl with the same pulse time are shown in Figure 13b. As expected, the significant lack of influence of the charging current on the current signal is observed at higher support. Another important point is that the reduction currents start to dominate at around 350 mV for both electrodes. This is a very important point when considering the electrocatalytic properties of SWCNTs. However, when the potential is further reduced, the current signals split in the same way as observed for the dependence of RPV signals on concentration. Accordingly, the difference in the cathodic signal can be attributed to the difference in Fe(III) concentration near the GCE.

This difference arises from the fact that during the electrochemical generation of Fe(III) at the bare GCE, the Fe(III) can diffuse into the bulk of the solution, decreasing the amount of Fe(III) near the GCE. On the contrary, in the case of the SWCNTs electrode, the generated Fe(III) remains trapped in the vicinity of the GCE, as well as in the porous SWCNTs layer. Finally, at potentials more negative than 125 mV, a significant change in the currents can now be observed, regardless of the type of electrode. With respect to the small *t*_p_, this is not surprising since similar current values were collected on a time scale where diffusion is mitigated. Accordingly, the collected signal currents are independent of a more negative potential.

Figure 13c,d show a comparison of the RPVs presented above, but for the same electrode in different support concentrations. An interesting observation was made for the GCE in 0.001 M KCl, where no charging currents at anodic potentials were observed at the measured pulse time. This strongly suggests that in the case of the polished GCE, the relaxation of the double layer due to a small potential perturbation is negligible compared with the SWCNTs electrode, due to its smaller charging electrode surface. When the base potential is applied (0.6 V for 0.45 s), the double layer is completely formed. Applying a short pulse with a potential step of 25 mV during the anodic branch of the RPV does not lead to any measurable relaxation current in the case of GCE due to their low total capacitance. The opposite is the case with the SWCNTs electrode.

Significant changes were obtained by increasing the pulse time (*t*_p_ = 0.4 s), as the contribution of the relaxation and charging currents in the total signal becomes negligible regardless of the support ionic strength. Accordingly, the anodic part of the RPVs is characterized by an anodic “plateau”, even for SWCNTs in 0.001 M KCl (see Figure 14c). Since the system has enough time to relax the double layer, the current is only the consequence of the oxidation of the remaining Fe(II).

In addition, the cathodic current in the case of GCE starts at a more negative potential than in SWCNTs (Figure 14a). In explaining the presented results, we rely on the influence of the applied surface electrode potential on the composition and potential profile across the diffuse or Gouy–Chapman (GC) layer in the Gouy–Chapman–Stern model of the double layer, and consequently, on Frumkin effects [30,31]. Accordingly, ion condensation in the GC layer near the GCE is more pronounced in the SWCNTs electrode due to the restricted electrolyte volume. Furthermore, this effect is more pronounced with increasing applied potential. As a consequence, the effective potential drop near the GCE can be considered somewhat large in the case of the modified electrode. As a result, the reduction starts at a slightly lower cathodic potential in the case of the SWCNTs electrode. In combination with the previously described diffusion of the electrochemically generated Fe(III) toward the bulk solution, the Fe(III) in the vicinity of the GCE for the SWCNTs electrode can be considered higher than in the case of the bare GCE. All this contributes to the shape of the recorded voltammograms. For instance, in the case of the SWCNTs electrode, the voltammograms show the maximum cathodic current at a lower cathodic potential than in the case of the GCE. Furthermore, the application of a higher effective potential and a long *t*_p_ results in electrolyte depletion near the GCE, which leads to the recording of a lower cathodic current than in the previous steps. As a result, the RPVs show a peak shape. This is not observed in the case of GCE, due to the lower effective potential that leads to a slower expansion of the diffusion layer. Moreover, the diffusion layer can be considered wider in the case of GCE than in the case of SWCNTs electrode (considering that the thickness of the diffusion layer can vary from 1 to 100 μm) [39]. This is consistent with the micrograph presented in Appendix A (the largest pore size is around 200 nm)

At higher support (Figure 14b), due to a similar extent of the compression of the GC layer as a consequence, the cathodic currents start to dominate at the same potential (280 mV) for both electrodes, indicating that no electrocatalytic effects can be attributed to the SWCNTs. Interestingly, no difference in peak potential is observed between the SWCNTs electrode and the GCE in 0.1 M KCl. This clearly indicates that the reduction takes place on the same surface but at different amounts of Fe(III).

Figure 14c shows the comparison of the RPVs of SWCNTs electrodes recorded in different ionic strengths of KCl at *t*_p_ = 0.4 s. There is no significant difference in the RPVs up to a potential of 200 mV. Following the previous discussion, this is the consequence of the formation of a compressed and ion-condensed GC layer near the GCE. Accordingly, the potential drop can be considered similar to that in 0.1 M KCl. However, at a more negative potential, the more pronounced cathodic currents were recorded for RPV in 0.001 M?! Based on *t*_p_ = 0.4 s, the assumption of complete reduction of Fe(III) near the electrode seems reasonable, especially considering the results of the DPSCAs analysis. Accordingly, during the base potential (i.e., during 0.1 s of Fe(II) oxidation), the formation of a lower amount of Fe(III) is expected in the case of 0.001 M KCl. This excludes the possibility of an influence of the Fe(III) concentration on this phenomenon and can be attributed to the delayed behavior (i.e., rate) of the faradaic reaction in low support concentration. Also, during the cathodic pulse time, at a more cathodic potential, compression of the GC layer simultaneously occurs. As a result, the maximum effective potential difference, in the plane of the electron transfer, is formed later than in the case of 0.1 M KCl. As the GC layer is compressed, the effective potential drop also increases, which increases the reduction rate. However, the elapsed pulse time and the cathodic current are being collected at a higher value. The time consumed in compressing the GC layer results in the peak of the cathodic current occurring at a slightly higher cathodic potential (−12 mV relative to the peak observed in 0.1 M KCl). We find that this is a low cost considering that the signal is more than 50% higher in the 0.001 M case. Here, we must emphasize that the presented discussion is based on the assumption that the plane of the electron transfer, where quantum tunneling occurs, can reach 20 Å [40].

### 2.4. Cyclic Voltammetry—Discussion

Based on all that was discussed above, the electrochemical behavior of Fe(II) on SWCNTs as well as on GCE can be easily elucidated. The most interesting observation of the CVs is that for GCE in 0.001 M KCl. As shown in Figure 1a, the dependence of the anodic current peak on the scan rate shows a behavior suggesting that the reaction proceeds under diffusion control. Small discrepancies can be observed for CVs recorded in 0.001 M KCl at higher scan rates. However, differences in the potential of the current peaks vs. scan rate can be observed in the whole range. The latter is to be expected if we rely on the discussion of the RPV results with low support. However, the cathodic current peaks do not follow this trend (Figure 1c). Consequently, the cathodic behavior cannot be rationalized only by the low support and ion condensation arguments, but rather by the presence of electroactive species in the bulk solution.

Due to the technical principles of staircase voltammetry, the GC layer can be considered compressed near the potentials of the current peaks at scan rates up to 100 mV s^−1^ (Figure 1a) when measurements are performed in 0.001 M KCl. Another aspect relevant to this discussion concerns the spreading of the electroactive species towards the bulk. As already mentioned, this is particularly important for GCE. When the diffusion layer is “reached” with electroactive species, the spreading (expansion or spontaneous diffusion) of the electroactive species towards the bulk during the half-cycle is less pronounced. This happens in the case of the anodic branch of the CV because Fe(II) is present in the bulk. Of course, if the scan rate is fast enough, this spreading should be slower. Accordingly, the effective potential near the peak potential is lower, resulting in the lower current being collected.

At a slow scan rate (12.5 mV s^−1^), the step time during staircase voltammetry is sufficient to establish diffusion conditions independent of the ionic strength of KCl (see explanation in the RPV experiments), which leads to the formation of the same current peak heights (when comparing cathodic and anodic). In fact, no differences were observed between peak heights when changing KCl concentrations, as no significant influence of GC layer compression is expected.

A further increase in the scan rates results in the significant splitting of the cathodic current peak heights when compared with the anodic ones. In fact, at high scan rates, the cathodic branch of CVs is characterized by constant current values independent of the potential. This may be a consequence of the following:The shortest step time, based on the fact that the potentiostat does not change the step potential upon changes in the scan rate (but obviously the step time).Constant Fe(III) flux.

The phenomena during cathodic scanning related to the compression of the GC layer can be considered identical to those in the anodic branch. Accordingly, the first claim is directly related to the effective potential (i.e., the phenomena of GC layer compression) and should also be observed in the anodic branch of CVs at 0.001 M KCl. Since this was not observed, it can be excluded from the discussion.

In contrast to the anodic process, the concentration gradient of the electrochemically generated Fe(III) during the cathodic branch is directed towards the bulk (c(Fe(III)_bulk_ = 0), which indicates an uneven spatial distribution of Fe(III) in the double layer. This indicates that the diffusion zone is larger than in the case of the anodic branch, when the electroactive species is present in the bulk of the solution. At this point, we must emphasize that when analyzing the charge transferred during oxidation (area under the peak), almost equal charges were transferred when the measurements were performed at the same scan rate, regardless of the concentration of the support. This suggests that equal concentration of Fe(III) is present in the system at the beginning of the cathodic branch of CV.

Accordingly, during cathodic scanning in the case of a kinetically limited reaction, a steady state is to be expected due to the wider diffusion zone (and thus a longer diffusion path), as observed in the case of 0.001 M KCl. As a result, lower cathodic currents were recorded at higher scan rates, followed by their independence upon potential (see Appendix A).

This behavior is characterized by the slope of log(*i*) vs. log(υ) of 0.16. In the case of the higher support, the depletion of the diffusion zone is directly correlated with the increase in the cathodic potential due to the high effective potential. Accordingly, no steady state condition can be observed, and the peak shape is obtained for the cathodic branch even at the highest scan rate. This is manifested by the slope of the log(i) vs. log(υ) slope of 0.42. Here we must emphasize that the explanation is based on the fact that the potentiostat changes do not change the step potential with the changes in scan rates (but obviously the step time).

Finally, the analysis of the experimental data extracted from the CVs recorded at the SWCNTs (Appendix A), dependent on the scan rate (Figure 2), confirms all assumptions and explains the phenomena based on the DPSCA, RPV, and CV on the GCE electrode results. Since the SWCNTs act as a blocking electrode not only towards the GCE electrode but also as a barrier to the diffusion of Fe(III) into the bulk of the solution, a thinner diffusion zone results. Consequently, the dependence of the cathodic and anodic current peaks on the scan rate suggests a diffusion-controlled reaction at all ionic strengths of KCl, indicating that the diffusion layer is depleted in SWCNTs at the same rate. This is possible due to the limited amount of Fe(III) trapped in the small volume near the GCE, but it is not the case for the anodic branch, as the expansion of the diffusion layer is highly influenced by the presence of Fe(II) in the bulk and correspondingly in the porous layer of the SWCNTs. It is also important to point out the unusual changes in the position of the anodic current peaks when changing the scan rate, which are more pronounced when the KCl concentration is decreased. This inherently counterintuitive behavior is predicted by the simulation of the voltammetric behavior of electrodes covered with porous inactive materials by Kathelon et al. [41]. This behavior is more pronounced at low support concentration and slow scan rates, as it fits to the time frame of the processes that occur in SWCNTs during cycling.

## 3. Materials and Methods

### 3.1. Chemicals and Solutions

Single-walled carbon nanotubes (SWCNTs) with a diameter of 0.7–0.9 nm and a carbon content of ≥93% (as SW) were purchased from Sigma-Aldrich (St. Louis, MI, USA). Alumina powder with particle sizes of 1 μm and 0.05 μm was used to polish the GCE, which was purchased from Leco (St. Joseph, MI, USA). All chemicals used were of analytical grade, and all solutions were prepared with ultrapure water (18 MΩ cm). Sulfuric acid (*w* = 98%), dimethylformamide, potassium hexacyanoferrate(II), and potassium chloride were supplied by Kemika (Zagreb, Croatia). A 0.1 M stock solution of KCl was used to prepare working solutions with final concentrations of 0.001 M and 0.01 M. Solutions of potassium hexacyanoferrate(II) were prepared daily by dissolving the appropriate amount of K_4_[Fe(CN)_6_] × 3H_2_O in KCl solutions in order to obtain aimed concentrations. For simplicity, throughout the article, K_4_[Fe(CN)_6_] and electrochemically generated K_3_[Fe(CN)_6_] are marked as Fe(II) and Fe(III), respectively.

### 3.2. Apparatus

Cyclic voltammetry (CV), reverse pulse voltammetry (RPV), and double potential step chronoamperometry (DPSCA) were performed with a potentiostat (Autolab PGSTAT 302N, Aurora, CO, USA) connected to a PC and controlled by GPES 4.9 software (Eco Chemie, Utrecht, The Netherlands). A conventional electrochemical cell with three electrodes was used, consisting of a Radiometer XR300 Ag/AgCl reference electrode (sat. KCl + sat. AgCl) and a platinum plate as the auxiliary electrode. Homemade working GCE was prepared by sealing glassy carbon rods (K-type, diameter 2 mm, Sigradur, HTW, Thierhaupten, Germany) in the epoxy resin. An ultrasonic bath (Bandelin SONOREX, Berlin, Germany, model RK 31) was used to remove impurities from the electrode surface, while a second ultrasonic bath (Bandelin SONOREX, model RK 103H) was used to homogenize the SWCNTs. Morphological characterization of the electrode surface was performed using a Zeiss Sigma500 VP scanning electron microscope with Zeiss SmartSEM v05.06 imaging software.

### 3.3. Procedures

#### 3.3.1. Electrode Preparations

Prior to surface modification, the glassy carbon electrode (GCE, *A* = 0.0314 cm^2^) was thoroughly polished with alumina powder of 1 μm and 0.05 μm particle size. The electrode was then ultrasonically cleaned in redistilled water, rinsed with ethanol and redistilled water for one minute, dried under a stream of nitrogen, and finally electrochemically cleaned. Electrochemical cleaning was performed in 0.5 M sulfuric acid using cyclic voltammetry (CV) over a potential range of −0.2 V to +1.2 V at a scan rate of 100 mV s^−1^. After cleaning, the electrode was rinsed again with redistilled water.

SWCNTs powder was dispersed in dimethylformamide (DMF) in an ultrasonic bath for 24 h to produce a suspension of 0.5 mg mL^−1^. The SWCNTs modified GCE (SWCNTs electrode) was obtained by dropping 1.5 μL of the SWCNTs suspension onto the pretreated GCE surface. The solvent was evaporated for one hour at room temperature. The prepared electrode was immersed in pure ethanol for 3 s and then rinsed in water. A new SWCNTs-modified layer was prepared for each series of measurements. All electrochemical measurements were repeated three times. No significant change in electrochemical behavior and signals was observed when comparing the measurements.

In the case of SWCNTs modified GCE grounded with paper 3000 (GCE-3000 SWCNT), the electrode was prepared by grinding the glassy carbon surface in one direction with 3000-grit abrasive paper. The procedure for cleaning and drop-casting of SWCNTs was the same as for the polished electrode.

#### 3.3.2. Cyclic Voltammetry

Cyclic voltammetry (CV) measurements were performed within a potential window from −0.1 V to +0.6 V vs. Ag/AgCl. Cyclic voltammograms (CVs) were recorded in the presence of 0.5 mM Fe(II) in different KCl concentrations (0.001 M, 0.01 M, and 0.1 M) and scan rates from 12 mV s^−1^ to 400 mV s^−1^.

#### 3.3.3. Double Potential Step Chronoamperometry

The measurements were carried out by recording a double potential step chronoamperograms (DPSCAs) in a solution of Fe(II) at +0.50 V vs. Ag/AgCl, followed by a switch potential to +0.03 V vs. Ag/AgCl. Pulse time for each potential was 10 s. It should be noted that for analysis, only the faradaic currents were considered.

#### 3.3.4. Reverse Pulse Voltammetry

Reverse pulse voltammograms (RPVs) were recorded in a solution containing 0.5 mM Fe(II) in either 0.001 M or 0.1 M KCl between +0.60 V and −0.10 V vs. Ag/AgCl, at scan rate of 50 mV s^−1^, interval time *τ* = 0.5 s and base potential of +0.60 V. Two sets of measurements was carried out: at pulse time *t*_p_ = 0.05 s and at *t*_p_ = 0.4 s.

## 4. Conclusions

By analyzing the cathodic branch of double-potential step chronoamperograms under different experimental conditions, a novel criterion is proposed to distinguish two mass transport domains: one within the porous layer and another that also includes changes in the flux in the bulk of the solution. Based on this criterion, the points at which mass transport transitions from one region to the other are determined to be between 1.4 and 2.2 s, depending on the ionic strength of the support. This is also recognized as slope values around −0.65 in log(*I*) vs. log(*t*) dependence. These data strongly indicate the inactivity of the SWCNTs. However, the influence of SWCNTs is crucial as they act as a barrier for diffusion into the solution and through their EDLC behavior.

Reverse pulse voltammetry shows that the experimental parameters play a crucial role in obtaining the cathodic signal. By choosing suitable parameters such as pulse and interval time, the current signal can be significantly improved. Surprisingly, a higher cathodic signal in support of the lower ionic strength was obtained when the measurements were performed with a porous SWCNTs modified GCE. This was attributed to the later establishment of the maximum effective potential. Also, it is followed by a negligible shift in potential compared with the fully supported condition.

The cyclic voltammetry data indicate that the reactions (both anodic and cathodic) proceed under diffusion control when only Fe(II) is present in solution. The discrepancies of the cathodic behavior from the values predicted by the Randles–Sevcik expression were found in the case of low support at the GCE electrode. This discrepancy was mainly attributed to the expansion of the diffusion layer, although the influence of ion condensation in the diffuse layer cannot be neglected.

In contrast, the parameters determined for the electrode modified with SWCNTs suggest that this effect is absent, as the slopes of log(*I*) versus log(*t*) indicate diffusion-controlled behavior regardless of the ionic strength of the support. This is attributed to the enhanced ion condensation in a small volume in the vicinity of the GCE. As a consequence of the small volume of the electrolyte and enhanced ion condensation, increased diffusion fluxes are observed at all ionic strengths of the support.

The presented results may have an impact on the elucidation and further improvement and modeling of the electrochemical system based on porous materials, both conductive and non-conductive, not only in the field of electrochemical sensing, but also in various energy storage systems such as EDLC, pseudo-capacitive, and other electrochemical devices. Moreover, the presented analysis and proposed criteria can be considered as an alternative method to the recently reported single-entity approach (summarized in ref. [15]) to remove a doubt and to elucidate the role of the carbon nanotubes or even other porous materials (such as metal–organic frameworks, modified CNT systems, etc.) used in various electrochemical devices.

## Figures and Tables

**Figure 1 ijms-26-08262-f001:**
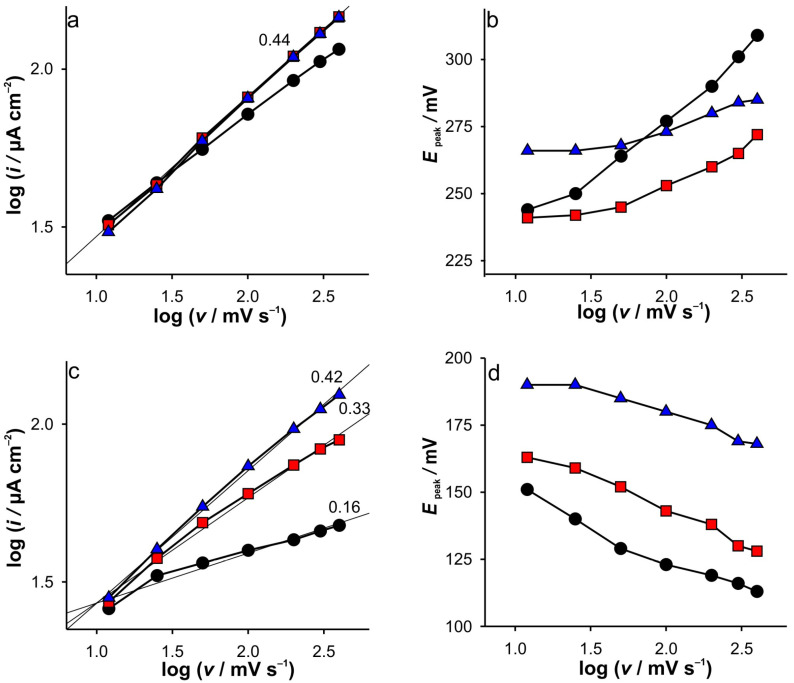
(**a**) Dependences of log(*i*) vs. log(*υ*) and (**b**) *E*_peak_ vs. log(υ) for anodic branch of CVs; (**c**) dependences of log(*i*) vs. log(*υ*) and (**d**) *E*_peak_ vs. log(υ) for cathodic branch of CVs. The results are obtained for GCE at KCl concentrations: (●) 0.001 M; (■) 0.01 M; (▲) 0.1 M.

**Figure 2 ijms-26-08262-f002:**
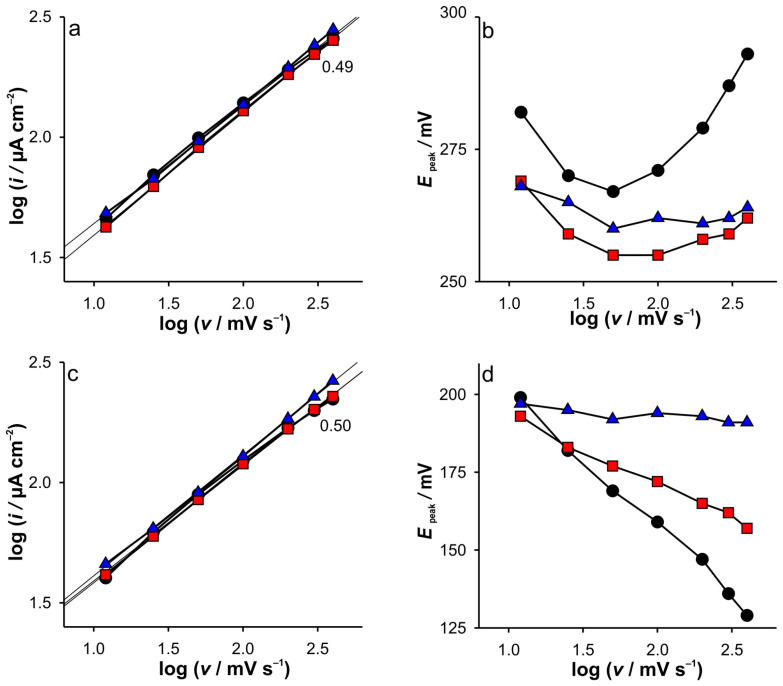
(**a**) Dependences of log(*i*) vs. log(*υ*) and (**b**) *E*_peak_ vs. log(υ) for anodic branch of CVs; (**c**) Dependences of log(*i*) vs. log(*υ*) and (**d**) *E*_peak_ vs. log(υ) for cathodic branch of CVs. The results are obtained for SWCNTs electrode at KCl concentrations: (●) 0.001 M; (■) 0.01 M; (▲) 0.1 M.

**Figure 3 ijms-26-08262-f003:**
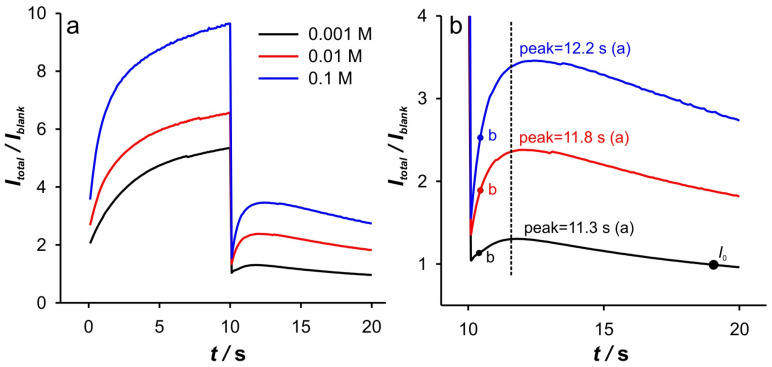
(**a**) Dependence of *I*_total_/*I*_blank_ vs. *t*, for the SWCNTs electrode, at 0.1 mM Fe(II) in different KCl concentrations. (**b**) Dependence of *I*_total_/*I*_blank_ vs. *t* for the cathodic branch presented in (**a**).

**Figure 4 ijms-26-08262-f004:**
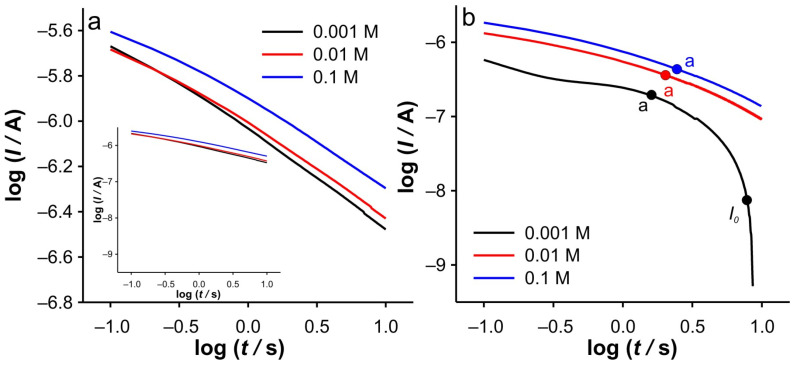
log(*I*) vs. log(*t*) dependencies for SWCNTs electrode, calculated from DPSCAs recorded at 0.1 mM of Fe(II) in different KCl concentrations for: (**a**) anodic branch of DPSCAs (Inset shows the same curves plotted using the scale of the cathodic branch); (**b**) cathodic branch of the DPSCAs. Note: The time values for points “a” and “*I*_0_” correspond to those determined in Figure 3b.

**Figure 5 ijms-26-08262-f005:**
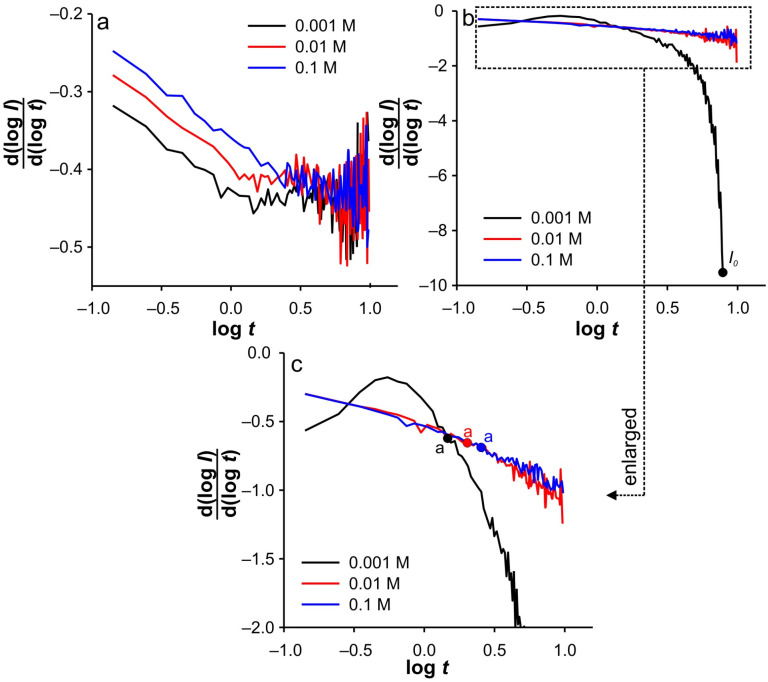
d(log(*I*))/d(log(*t*)) vs. log(*t*) dependencies for SWCNTs electrode at 0.1 mM of Fe(II) in different KCl concentrations for: (**a**) anodic branch of DPSCAs; (**b**) cathodic branch of DPSCA; (**c**) enlarged region for the cathodic branch of DPSCAs. Note: d(log(*I*))/d(log(*t*)) vs. log(*t*) dependencies are extracted from Figure 4, while the time values for points “a” and “*I*_0_” correspond to those determined in Figure 3b.

**Figure 6 ijms-26-08262-f006:**
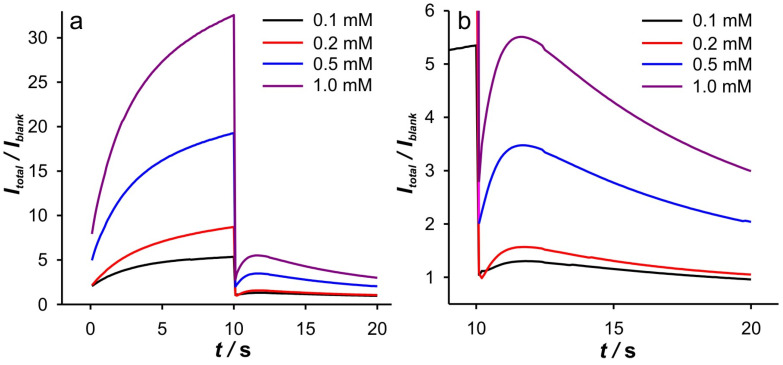
(**a**) Dependence of *I*_total_/*I*_blank_ vs. *t*, for the SWCNTs electrode, in 0.001 M KCl at different concentrations of Fe(II). (**b**) Dependence of *I*_total_/*I*_blank_ vs. *t* for the cathodic branch presented in (**a**).

**Figure 7 ijms-26-08262-f007:**
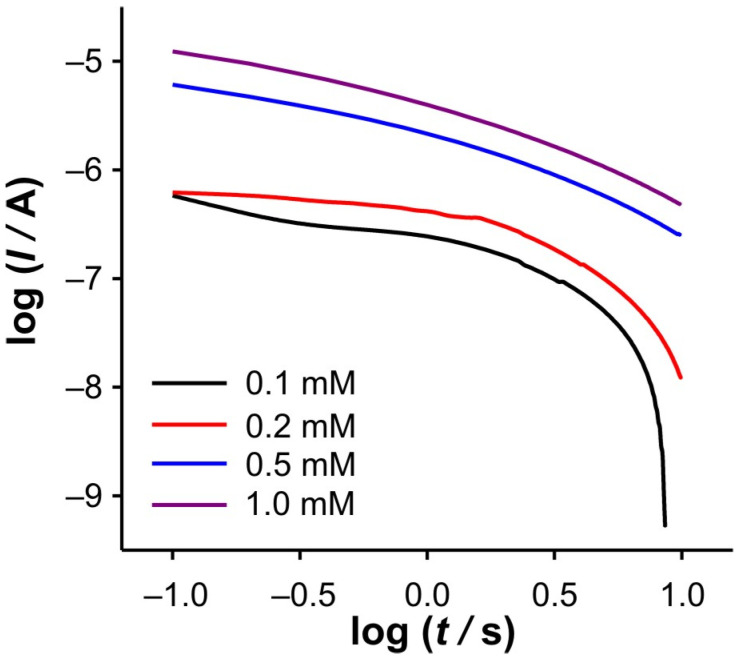
log(*I*) vs. log(*t*) dependencies for cathodic branch of DPSCAs for SWCNTs electrode. The data were calculated from the DPSCAs recorded in 0.001 M KCl at different concentration of Fe(II).

**Figure 8 ijms-26-08262-f008:**
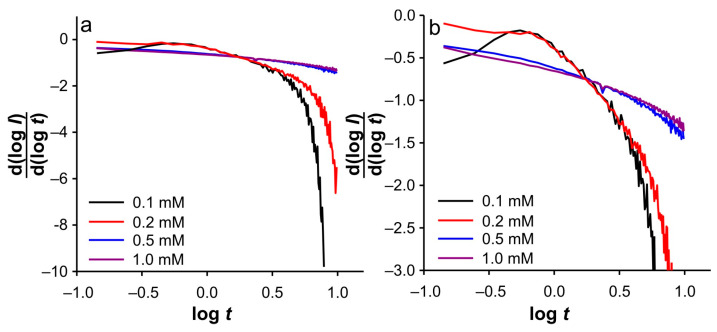
d(log(*I*))/d(log(*t*)) vs. log(*t*) dependencies for SWCNTs electrode in 0.001 M KCl at different concentration of Fe(II) for: (**a**) cathodic branch of DPSCAs; (**b**) enlarged region for the cathodic branch of DPSCAs. Note: d(log(*I*))/d(log(*t*)) vs. log(*t*) dependencies are extracted from Figure 7.

**Figure 9 ijms-26-08262-f009:**
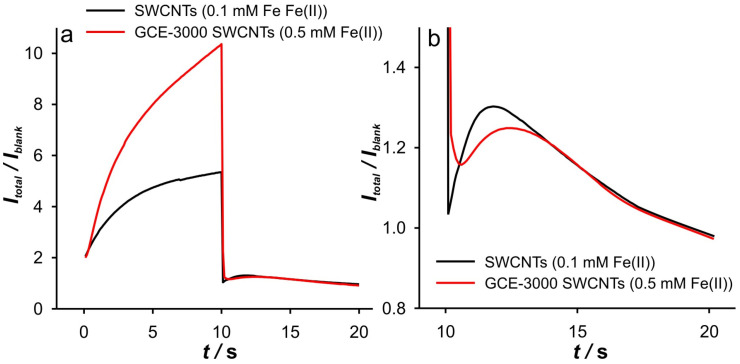
(**a**) Dependence of *I*_total_/*I*_blank_ vs. *t*, for different electrodes in 0.001 M KCl. (**b**) Dependence of *I*_total_/*I*_blank_ vs. *t* for the cathodic branch presented in (**a**). Note the difference in Fe(II) concentration.

**Figure 10 ijms-26-08262-f010:**
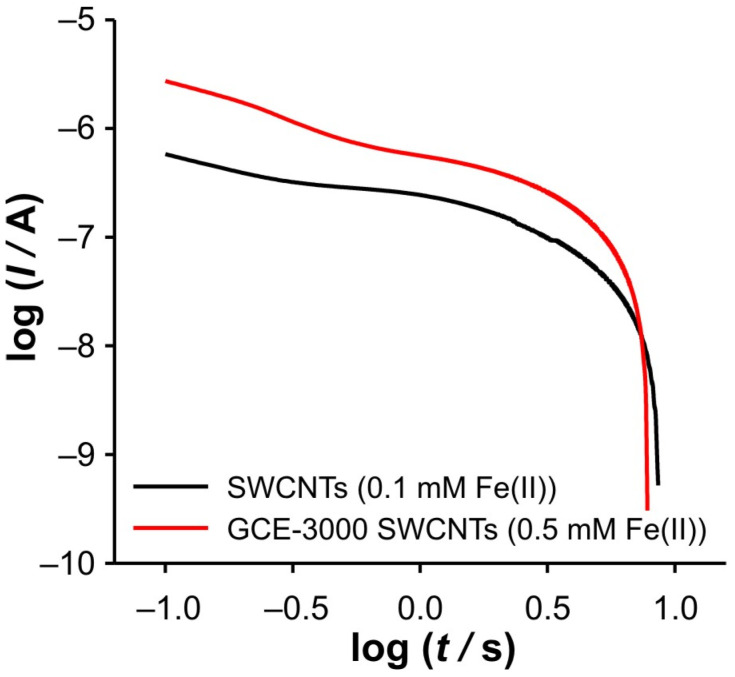
log(*I*) vs. log(*t*) dependencies for the cathodic branch of DPSCAs, for different electrodes in 0.001 M KCl. Note the difference in Fe(II) concentration.

**Figure 11 ijms-26-08262-f011:**
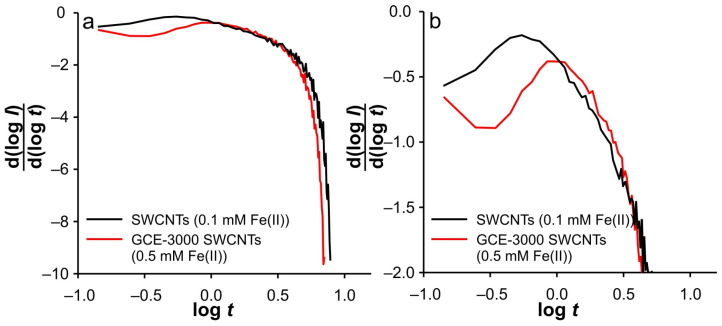
d(log(*I*))/d(log(*t*)) vs. log(*t*) dependencies for different electrodes in 0.001 M KCl for: (**a**) cathodic branch of DPSCAs; (**b**) enlarged region for the cathodic branch of DPSCAs. Note the difference in Fe(II) concentration. Note: d(log(*I*))/d(log(*t*)) vs. log(*t*) dependencies are extracted from Figure 10.

**Figure 12 ijms-26-08262-f012:**
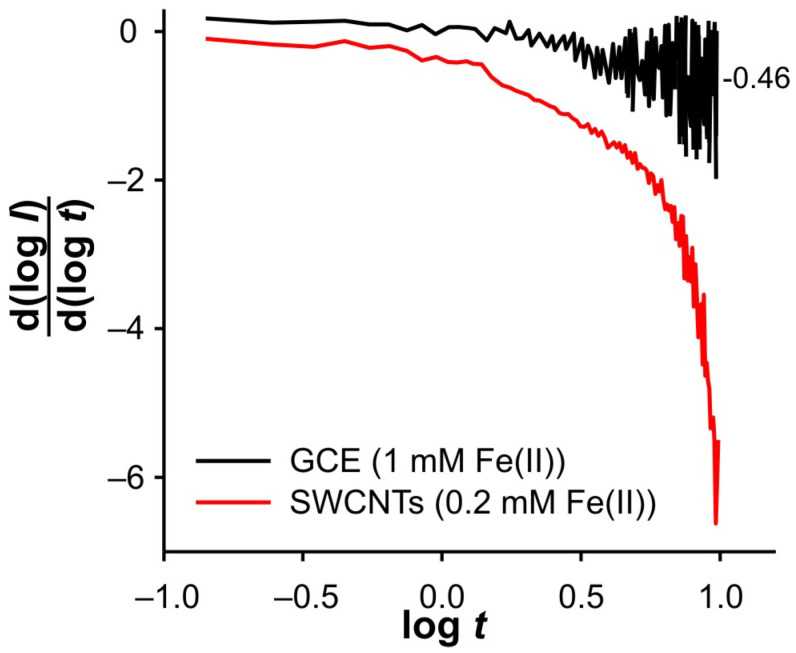
d(log(I))/d(log(t)) vs. log(t) dependencies for different electrodes in 0.001 M KCl for cathodic branch of DPSCAs. Note the difference in Fe(II) concentration.

**Figure 13 ijms-26-08262-f013:**
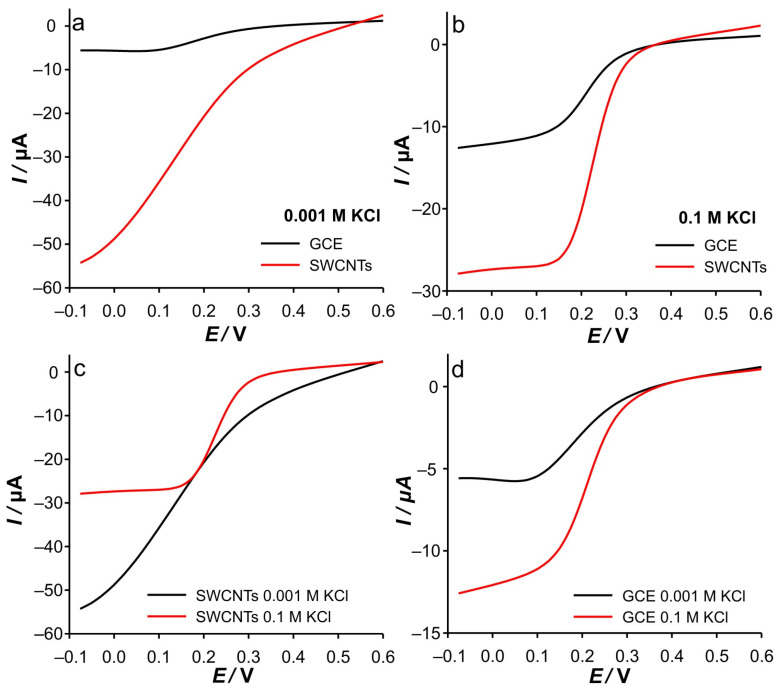
(**a**) RPVs recorded in 0.001 M KCl with different electrodes. (**b**) RPVs recorded in 0.1 M with different electrodes. (**c**) RPVs recorded with SWCNTs electrode in different KCl concentrations. (**d**) RPVs recorded with GCE in different KCl concentrations. All RPVs were recorded in 0.5 mM Fe(II) at a scan rate of 50 mV s^−1^, a pulse time *t*_p_ = 0.05 s, and an interval time *τ* = 0.5 s.

**Figure 14 ijms-26-08262-f014:**
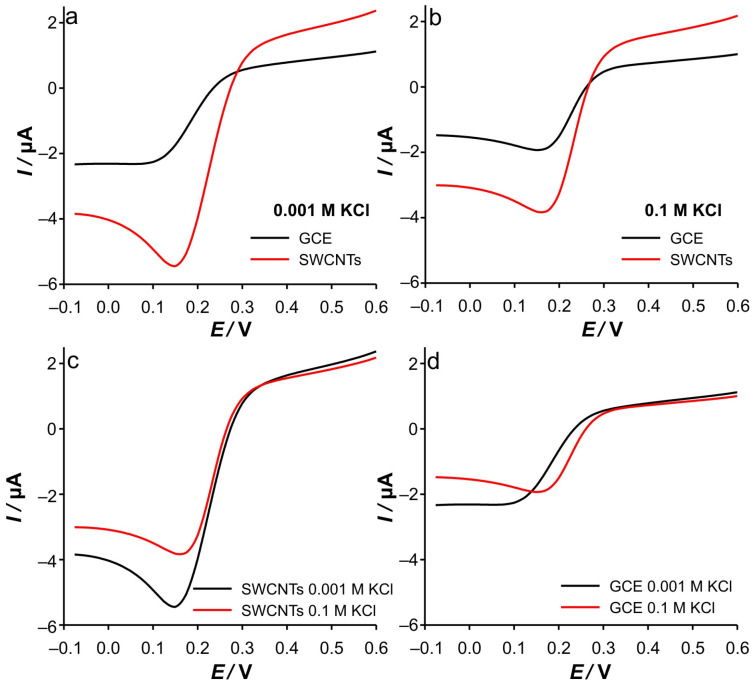
(**a**) RPVs recorded in 0.001 M KCl with different electrodes. (**b**) RPVs recorded in 0.1 M with different electrodes. (**c**) RPVs recorded with SWCNTs electrode in different KCl concentrations. (**d**) RPVs recorded with GCE in different KCl concentrations. All RPVs were recorded in 0.5 mM Fe(II) at a scan rate of 50 mV s^−1^, a pulse time *t*_p_ = 0.4 s, and an interval time *τ* = 0.5 s.

**Table 1 ijms-26-08262-t001:** Parameters of the points “a” (the boundary between different diffusion domains), extracted from Figure 3.

*c*(KCl)/M	Relative Cathodic Time */s	log(*t*)	Slope/d(log(*I*))/d(log(*t*))
0.001	1.4	0.16	−0.63
0.01	1.8	0.26	−0.64
0.1	2.2	0.34	−0.66

* Calculated as: total time of DPSC experiment—10 s.

## Data Availability

The data presented in this study are available within the content of this article.

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
