# Peer review of "A Fundamental Study on a Porous Carbon Nanotubes Macroelectrode in Weakly Supported Electrolyte: A Novel Criterion for Distinguishing Diffusion Domains"

_ijms, 2025, doi:10.3390/ijms26178262_

Round 1
Reviewer 1 Report
Comments and Suggestions for Authors
The manuscript presents a thorough investigation of the electrochemical behavior of SWCNT-modified electrodes in weakly supported electrolytes, with a focus on mass transport regimes and double-layer effects. I believe the study presents valuable insights into the electrochemical behavior of porous carbon nanotubes electrodes. However, there are several areas that require minor revisions to enhance the clarity and impact.
- The section on electrode preparation could benefit from more detailed descriptions of the cleaning and drop-casting processes. This will help other researchers to replicate your work more accurately.
- In the Materials and Methods section, please provide more information on the specific parameters used for the electrochemical measurements, such as the exact values of pulse times and potentials.
- The discussion section should include a more detailed comparison of your results with existing literature.
- Please elaborate on the implications of your findings for the development of new electrode materials and their potential applications in energy storage devices.
- There are a few grammatical errors and awkward phrasings throughout the manuscript.
- To strengthen the theoretical framework, the authors should cite:
1)Zhu Guangfei, Liu Rumeng, Wang Lifeng. Symmetry breaking and dynamic characteristics of post-buckling in bilayer van der Waals structures. International Journal of Solids and Structures. 2025, 309: 113190.
2)Zhu Guangfei, Liu Rumeng, Tang Chun, Wang Lifeng. Dynamic tuning of moire superlattice morphology by out-of-plane deformation. Applied Physical Letters. 2024, 124(17): 173508.
Reviewer 2 Report
Comments and Suggestions for Authors
This is an interesting and thorough study on the electrochemical behavior of porous SWCNT-modified electrodes in weakly supported electrolyte environments. Your use of multiple electrochemical techniques (CV, DPSCA, RPV) is appreciated, and the experimental data is comprehensive.
However, I have the following major concerns and suggestions:
1. Novelty Clarification
The proposed criterion (based on log(i)-log(t) slope) is interesting, but it appears mainly empirical. Please explain more clearly how it is different from existing models or approaches, particularly in comparison with your previous work (Ref. [15]).
2. Need for Theoretical/Simulation Support
Your main conclusions rely heavily on data interpretation. To strengthen your claims, I recommend incorporating numerical simulations (e.g., COMSOL) to model:
-Diffusion through the porous SWCNT layer.
-Electrochemical double-layer behavior and ion condensation.
-The transition between mass transport regimes.
This would significantly improve the robustness of the proposed criterion and support the observed slope threshold (~–0.65).
3. Discussion Streamlining
-Some parts of the manuscript, especially in Sections 2.2 and 2.3, are repetitive. Please consider merging or simplifying these sections for clarity.
-Some figures (e.g., Fig. 5c, 8b, 11b) are overly complex. Adding simplified schematics or annotations may help.
Overall, I believe your work can make a meaningful contribution after addressing the above points.
Reviewer 3 Report
Comments and Suggestions for Authors
Marijo Buzuk, in the manuscript entitled "A Fundamental Study on Porous Carbon Nanotubes Macroelectrode in Weakly Supported Electrolyte: A Novel Criterion for Distinguishing Diffusion Domains," presented a new approach to elucidate the phenomena that occur within a porous single-walled carbon nanotubes (SWCNTs) modified glassy carbon electrode (GCE) and that influence the electrochemical behavior of the modified electrode. They showed the role of the porous SWCNTs layer in preventing the expansion of the double-layer as well as in the process of ion condensation in the Gouy-Chapman layer. However, some of the notes listed below were observed:
- The manuscript contains many typos and spelling mistakes. They should be corrected.
- It would be better if the authors provided the discussion of the two important points presented in section 2.1. “Cyclic voltammetry (CV) study”, directly after the results and not in section 2.4.
- The authors wrote, “This method was chosen due to our focus on studying the cathodic reaction, as this reaction should have less degrees of freedom as it is independent on the concentration of electroactive species in the bulk.”. This point needs more illustration.
- In general, the standard deviations are absent. The authors should clarify how many samples they prepared? And how many times did they repeat their tests?
- The authors wrote, “However, from this analysis it is very difficult to distinguish when the SWCNTs layer is depleted of Fe(III)”. This point needs more scientific investigations.
- The authors wrote, “However, in contrast to the reported simulation, in our case, a rapid depletion in Fe(III) leads to a rapid decline of the slope with time. This happens due to the fact that it can be assumed that concentration of the Fe(III) in bulk of solution is zero.” Why is the concentration of the Fe(III) in the bulk of the solution zero?
- Any chemical or physical characterization of the prepared electrode based on SWCNTs is absent.
- The main question is: Can the suggested criterion be considered general, and will it be convenient for all electrodes based on CNTs, or is it just related to the studied SWCNTs?
Round 2
Reviewer 2 Report
Comments and Suggestions for Authors
The authors have addressed almost all of the issues raised by the reviewers. Therefore, I recommend that this manuscript be accepted for publication in the International Journal of Molecular Sciences after minor revision.
-
Figure 2 should be modified by changing the colors representing the KCl concentrations (0.001 M, 0.01 M, and 0.1 M). As currently presented, it is difficult to distinguish the overlapping data.
Author Response
Figure 2 is modified according to your suggestion.
Thank you for your efforts to improve our article.
Best regards
Reviewer 3 Report
Comments and Suggestions for Authors
The authors made acceptable improvements to the manuscript.
Author Response
Thank you for your suggestions that significantly improved our article.
Best regards